Vegetative cell wall protein OsGP1 regulates cell wall mediated soda saline-alkali stress in rice

Zhu Fengjin 1
Cheng Huihui 1
Guo Jianan 1
Bai Shuomeng 2
Liu Ziang 3
Huang Chunxi 3
Shen Jiayi 3
Wang Kai 3
http://orcid.org/0000-0002-4877-9856 Yang Chengjun 3
http://orcid.org/0000-0002-9438-1427 Guan Qingjie 1 guanqingjie@nefu.edu.cn
1 Key Laboratory of Saline-Alkali Vegetation Ecology Restoration, Ministry of Education, College of Life Sciences, Northeast Forestry University , Harbin, Heilongjiang Province , China
2 Aulin College, Northeast Forestry University , Harbin, Heilongjiang Province , China
3 College of Forestry, Northeast Forestry University , Harbin, Heilongjiang Province , China
Mora-Montes Héctor
Electronic publication date: 2024 Feb 28
Publication date: 2024
Volume: 12
Electronic Location ID: e16790
Received 2023 Apr 20; Accepted 2023 Dec 21
Copyright: © 2024 Zhu et al.
Copyright year: 2024
Copyright holder: Zhu et al.
License: This is an open access article distributed under the terms of the Creative Commons Attribution License, which permits unrestricted use, distribution, reproduction and adaptation in any medium and for any purpose provided that it is properly attributed. For attribution, the original author(s), title, publication source (PeerJ) and either DOI or URL of the article must be cited.
License URL: https://creativecommons.org/licenses/by/4.0/

Keywords: Rice, Gene cloning, OsGP1, Genetic transformation, Abiotic stress

Funding: Science and Technology Fundamental Resources Investigation Special Project 2021FY100403 National Natural Science Foundation of China 32171989 Key Laboratory of Saline-alkali Vegetation Ecology Restoration (Northeast Forestry University) Ministry of Education 0924210201 This work was supported by the Science and Technology Fundamental Resources Investigation Special Project (2021FY100403), the National Natural Science Foundation of China (32171989) and financial from Key Laboratory of Saline-alkali Vegetation Ecology Restoration (Northeast Forestry University), Ministry of Education (0924210201). The funders had no role in study design, data collection and analysis, decision to publish, or preparation of the manuscript.

==============================
Plant growth and development are inhibited by the high levels of ions and pH due to soda saline-alkali soil, and the cell wall serves as a crucial barrier against external stresses in plant cells. Proteins in the cell wall play important roles in plant cell growth, morphogenesis, pathogen infection and environmental response. In the current study, the full-length coding sequence of the vegetative cell wall protein gene OsGP1 was characterized from Lj11 (Oryza sativa longjing11), it contained 660 bp nucleotides encoding 219 amino acids. Protein-protein interaction network analysis revealed possible interaction between CESA1, TUBB8, and OsJ_01535 proteins, which are related to plant growth and cell wall synthesis. OsGP1 was found to be localized in the cell membrane and cell wall. Furthermore, overexpression of OsGP1 leads to increase in plant height and fresh weight, showing enhanced resistance to saline-alkali stress. The ROS (reactive oxygen species) scavengers were regulated by OsGP1 protein, peroxidase and superoxide dismutase activities were significantly higher, while malondialdehyde was lower in the overexpression line under stress. These results suggest that OsGP1 improves saline-alkali stress tolerance of rice possibly through cell wall-mediated intracellular environmental homeostasis.

Introduction

The Songnen Plain of China is one of the three most concentrated saline-alkali lands in the world (Li et al., 2003), and it is an important reserve resource of cultivated land in China (Jiang et al., 2019). The salt in the soil of Songnen Plain mainly exists in the form of alkaline salts (NaHCO3 and Na2CO3), with alarmingly high concentration of salt and pH. Rice is one of the four major cereal crops (Zhang et al., 2003), the development of saline-alkali resistance in rice is an important research topic for improving the quality and efficiency of rice plantation and strengthening food security. Soda saline-alkali soil causes damage to plants mainly because of the high levels of Na+, CO32−, HCO3−, and extreme alkaline conditions (Wang et al., 2022a). Salt stress inhibits rice seed germination, seedling growth, and reproductive development (Li et al., 2003), leading to a decrease in leaf area, stalk, stem diameter, root activity, nutrient synthesis, accumulation, and transport, and interferes with young spike differentiation in reducing its effective spike number (Wang et al., 2022b). Salt-alkali stress also leads to various physiological and molecular changes and hinders plant growth by inhibiting photosynthesis, thereby reducing available resources and inhibiting cell division and expansion. Plants have evolved many biochemical and molecular mechanisms to cope with saline-alkali stress (Pastori & Foyer, 2002), mainly including ion-selective absorption, accumulation of osmotic adjustment substances, and scavenging of reactive oxygen species (ROS) (Liang et al., 2018). Salt stress leads to a large influx of Na+ into plant cells to induce ion toxicity, resulting in an imbalance of intracellular ion homeostasis, plants can promote osmotic balance at the cellular level through the synthesis of soluble sugars, proline, and other substances. ROS accumulates because of salt stress (Vaidyanathan et al., 2003), and plants scavenge excess ROS by producing superoxide dismutase (SOD), peroxidase (POD), ascorbate peroxidase (APX), and catalase (CAT) to avoid internal damage.

When plants are exposed to harsh climatic condition, the first to sense the stress is the cell wall. Plant cell walls surround cells and provide external protection and intercellular communication, and they are mainly composed of polysaccharides (cellulose, hemicelluloses, and pectins), polymers such as lignin, and a small amount of cell wall proteins (CWPs) (Jamet & Dunand, 2020). CWPs are divided into nine functional classes, including proteins acting on carbohydrates, oxidoreductases, proteases, proteins related to lipid metabolism, proteins possibly involved in signalling, proteins with predicted interaction domains, miscellaneous proteins, proteins of unknown function, and structural proteins (Calderan-Rodrigues et al., 2019). CWPs are major players in cell wall remodelling and signalling and play an important role in plant cell growth and development, morphogenesis, pathogen infection, and environmental response. FERONIA (FER), a plasma-membrane-localized receptor kinase from Arabidopsis, is necessary to sense the defects of the cell wall. Sensing of the salinity-induced wall defects is possibly a direct consequence of the physical interaction between the extracellular domain of FER and pectin. FER-dependent signalling elicits cell-specific calcium transients that maintain cell-wall integrity during salt stress (Feng et al., 2018). Leucine-rich repeat extensins (LRXs) are chimeric proteins in the cell wall. LRXs bind rapid alkalinization factor (RALF) peptide hormones that modify cell wall expansion and directly interact with the transmembrane receptor FER (Herger et al., 2019). RALF in turn interacts with FER. LRXs, RALFs, and FER function as a module to transduce cell wall signals to regulate plant growth and salt stress tolerance (Zhao et al., 2018).

Hydroxyproline-rich glycoproteins (HRGPs) are a superfamily of CWPs. According to the ‘Hyp contiguity hypothesis’, contiguous and non-contiguous clustered Hyp residues are the sites attached by arabinoside and arabinogalactan polysaccharide, respectively (Ma & Zhao, 2010). HRGPs modification involves proline (Pro) hydroxylation and subsequent O-glycosylation on Hyp residues (Hunt et al., 2017). HRGPs consist of three members: hyperglycosylated arabinogalactan proteins (AGPs), moderately glycosylated extensins (EXTs), and lightly glycosylated proline-rich proteins (PRPs), which function in diverse aspects of plant growth and development (Showalter et al., 2010). A total of 162 HRGPs have been identified in Arabidopsis proteome, including 85 AGPs, 59 EXTs, and 18 PRPs (Allan et al., 2010). Numerous studies have shown that EXTs are involved in cell wall reinforcement in higher plants and in defence against pathogen attacks (Castilleux et al., 2018, 2021; Otulak-Kozieł, Kozieł & Lockhart, 2018). The cell wall of the unicellular green alga Chlamydomonas reinhardtii does not contain cellulose nor other polysaccharides, it consists only of an insoluble HRGP framework and several chaotrope-soluble, hydroxyproline-containing glycoproteins (Jürgen, Ronald & Johannes, 2009). The wall enveloping the vegetative and gametic cells (V/G wall) has a highly ordered structure including a chaotrope-soluble crystalline layer assembled with well-characterized HRGPs (Jeffrey & Ursula, 1992). The chaotrope-soluble cell wall glycoprotein GP1 is the only polypeptide with an even higher proportion of hydroxyproline occurring in vegetative C. reinhardtii cells, and is a constituent of the insoluble cell wall components (Jürgen, Ronald & Johannes, 2009).

The remodelling and signal transduction functions of CWPs play an important role in abiotic stresses such as high temperature (Pinski et al., 2021), high salt (Feng et al., 2018; Zhao et al., 2018), and nutrient deficiency (Wu et al., 2019; Ogden et al., 2018) in plants. A total of 270 CWPs have been identified in Oryza sativa (Calderan-Rodrigues et al., 2019), however, the biological functions of rice vegetative cell wall proteins (GP1) involved in salt alkalinity resistance and related signal transduction and protease mechanisms have not been reported. In the current study, the OsGP1 gene was cloned from rice and the functional site of OsGP1 protein was determined by subcellular localization. The genetic phenotypes of tolerance to soda saline-alkali stress in rice overexpressing OsGP1 and wild-type were compared to clarify the role of OsGP1 under saline alkaline stress. These results support the involvement of OsGP1 in the stress resistance mechanism of rice cell wall under soda saline-alkali stress.

Materials and Methods

Plant material

O. sativa longjing11 (Lj11) seeds were donated by the research group of Qingyun Bu, Northeast Institute of Geography and Agroecology, Chinese Academy of Sciences.

Soda saline–alkali soil eluent

The soda saline-alkali soil eluent (SAE) used for stress treatment was obtained as described by Wang, Takano & Liu (2018). The 0–10 cm soil of heavy alkali patch was collected from the Anda field experiment station (124°53′~125°55′E, 46°01′~47°01′N). The soil was dried, passed through a 5 mm × 5 mm sieve, and mixed well. Approximately 4 l of water was poured into 2 l of saline-alkali soil. The mixture was stirred well and left for 12 h (stirring once every 4 h). The mixture was filtered using filter paper to remove impurities, and experimental SAE was obtained. The different ratios of SAE required in the experiments are shown in Table 1.

Table 1 Configuration and characteristics of different ratios of soda saline-alkali soil eluent.

Leachate	Configuration method	EC (µS·cm−1)	pH	
SAE (Stock solution)	100 ml SAE	10,290 ± 36	9.65 ± 0.08	
H2O:SAE = 2:1	66 ml H2O + 33 ml SAE	3,903 ± 38	9.21 ± 0.06	
H2O:SAE = 3:1	75 ml H2O + 25 ml SAE	2,927 ± 35	9.08 ± 0.05	
H2O:SAE = 4:1	80 ml H2O + 20 ml SAE	2,530 ± 26	8.95 ± 0.06	
Control (H2O)	100 ml H2O	82.6 ± 0.1	6.98 ± 0.04	
Note: Independent triplicate measurements were averaged and the standard deviation (SD±) was calculated.

Gene cloning strategy

According to the sequence of Nipponbare OsGP1 gene (XM_015773690), specific primers OsGP1-F1, R2 (Table S1) were designed. The RNA of Lj11 was reverse-transcribed and the resulting cDNA was used as a template, and the target DNA was amplified by Blend-Taq DNA polymerase. The DNA was purified, inserted into the pMD18-T vector, and then transformed into Escherichia coli JM109. The pMD18-T-OsGP1 plasmid was identified by digestion with restriction endonuclease SalI/BamHI and sent for sequencing. The binary plant expression vector pGWB5-OsGP1 was constructed by Gateway technology. The open reading frame (ORF) of OsGP1 without the stop codon was amplified from the pMD18-T-OsGP1 plasmid previously verified by sequencing using specific primers OsGP1-F3, R4 (Table S1), ligated it into the entry vector pENTR/D-TOPO using the pENTR/D-TOPO Cloning Kit (Invitrogen, Carlsbad, CA, USA), and then moved to pGWB5 using Gateway LR Clonase II Enzyme Mix (Invitrogen, Carlsbad, CA, USA). The construction strategy created with Biorender (https://www.biorender.com/) is shown in Fig. S1A. The recombinant plasmids were transformed into E. coli TOP10 strain and positive colonies were selected and confirmed by PCR to obtain pGWB5-OsGP1 which carries a translational fusion of the GFP reporter gene at the C-terminus of OsGP1 driven by 35S promoter (35S::OsGP1-GFP). Details of the linkage between OsGP1 and GFP proteins are shown in Fig. S1B. The schematic diagram of the T-DNA insertion site in pGWB5-OsGP1 is shown in Fig. 1.

Figure 1 Schematic diagram of the T-DNA region in pGWB5-OsGP1 plasmid.

LB, T-DNA left border; RB, T-DNA right border.

Bioinformatics analysis

The nucleotide sequence obtained by sequencing was used as the comparison benchmark, and the ORF of OsGP1 was analysed based on NCBI (https://www.ncbi.nlm.nih.gov/). The physicochemical properties of the encoded protein were obtained using the ProtParam (https://web.expasy.org/protparam/). The online tool SOPMA (https://npsa-prabi.ibcp.fr/cgi-bin/npsa_automat.pl?page=npsa%20_sopma.html) was used to predict the protein’s secondary structure. The tertiary structure was predicted using SWISS-MODEL (https://swissmodel.expasy.org/). Conserved domains were analysed using the online website SMART (https://smart.embl.de). Signal peptides were analysed using SignalP-4.1 (https://services.healthtech.dtu.dk/services/SignalP-4.1/). The Locus ID (Os03g0852400) was found according to OsGP1 (XM_015773690). OsGP1 is located at the position of 35895654-35896833 (-strand) on chromosome 3. The sequence at 35896832-35898832 was taken as the OsGP1 promoter, the promoter elements in this sequence were analysed using Plant CARE (http://bioinformatics.psb.ugent.be/webtools/plantcare/html/) and mapped using Adobe Illustrator 2021. The sequences with high homology to OsGP1 were obtained by Blastp (NCBI) and aligned using Clustal Omega. The phylogenetic tree was constructed in MEGA6.0 by using the neighbour-joining (NJ) method (bootstrap value, 1,000). The subcellular location of OsGP1 was predicted by PSORT (https://psort.hgc.jp/).

Subcellular localization

To localize the OsGP1 protein, the pGWB5-OsGP1 plasmid was used. The recombinant plasmids were transformed into onion epidermal cells using a gene gun (GDS-80; Wealtec, Sparks, NV, USA) according to the manufacturer’s instructions, and the blank pGWB5 was transiently transformed as a control. After incubation at 22 °C for 16–24 h in the dark (Zhao et al., 2013), the onion epidermal cells were mounted on slides, and the GFP signal was observed under a fluorescence microscope (Axio Imager 2; ZEISS, Gottingen, Germany).

Rice transformation and selection

According to the rice transgenic method of Upadhyaya et al. (2000) and Toki et al. (2006), Lj11 seeds were dehulled and sterilized, and calli were induced on the medium added with 2,4-D. The pGWB5-OsGP1 plasmid was electrotransformed into Agrobacterium tumefaciens EHA105 and used to infect the rice calli. The infected rice calli was screened and differentiated on a hygromycin medium to obtain transgenic T0 generation lines, and the integration of OsGP1 was detected by PCR by using specific and vector primers B5-R (Table S1). The T3 generation transgenic rice was obtained by twice germination selection of 50 mg/l hygromycin medium. The relative expression of OsGP1 in T3 transgenic lines seedlings was detected by quantitative real-time PCR (qRT-PCR) in a fluorescent quantitative PCR instrument (Mx3000p; Agilent, Waldbronn, Germany). All expressions were normalized against the Os18sRNA gene. The primers used are listed in Table S1. The seeds of overexpressed T3 generation lines were collected for subsequent experiments.

Soda saline-alkaline stress tolerance analysis

To detect soda saline-alkali stress tolerance in rice overexpressing OsGP1, seeds of three independent homozygous transgenic lines (T3-#2, #4, and #5) of the T3 generation and the non-transgenic control (Lj11) were surface sterilized and cultivated in Hoagland nutrient solution at 28 °C with 16 h light and 8 h dark photoperiod. Three-leaf stage seedlings were used as treatment materials, and the compound salt mixed with sterilized water and SAE in different ratios was used as the stress treatment solution. The roots of each experimental line were soaked with different ratios of SAE, and the tolerance phenotypes and physiological indices were detected after 7 days. The detection method was based on Chen & Zhang (2016).

Data statistical analysis methods

Statistical analysis was carried out using IBM SPSS 26 for Windows, and one-way analysis of variance (ANOVA) with Tukey post hoc test was used for analysis. Statistical significance was defined as P ≤ 0.05.

Results

Cloning of OsGP1 gene

The PCR amplification of the OsGP1 gene resulted in a product of approximately 660 (Fig. 2A), then subsequent cloning process successfully inserted the amplified DNA into the pMD18-T vector, to generate the recombinant plasmid pMD18-T-OsGP1. The plasmid DNA was digested with SalI/BamHI, and electrophoresis results showed the DNA fragments were consistent with the PCR product (Fig. 2B). The plasmid was then confirmed by sequencing to obtain an identical nucleotide sequence comparison to the XM_015773690 predicted to be OsGP1 gene.

Figure 2 Electropherogram of OsGP1 DNA.

(A) Electropherogram of PCR amplification product from rice cDNA. M, Marker, DL2000; #1 and #2, PCR amplification products of OsGP1. (B) Recombinant pMD18-T-OsGP1 plasmid digested with SalI/BamHI. M: Marker, DL5000; #1, pMD18-T-OsGP1 plasmid; #2, pMD18-T-OsGP1 plasmids digested with SalI/BamHI.

Bioinformatics analysis of OsGP1

The ORF of OsGP1 contains 660 bp nucleotides and encodes 219 amino acids (Fig. 3A). The secondary structure of OsGP1 protein was predicted using SOPMA (Fig. 3B) and the tertiary structure was predicted by SWISS-MODEL (Fig. 3C), the results showed that it has α-helix and β-fold, accounting for 36.07% and 5.48%, respectively, while the remaining structures are mainly irregularly coiled with a few extended chains. Domain analysis showed that OsGP1 has three conserved domains and one transmembrane domain, of which a conserved domain at amino acid positions 173 to 199 overlaps with the transmembrane domain at 178 to 197 and is not shown (Fig. 3D). According to the signal peptide prediction results of SignalP (Fig. 3E), OsGP1 has a site at the 26th amino acid position that can be recognized and digested by signal peptidase. Hence, a possible signal peptide structure was predicted between the 1st and 25th amino acid sites. The protein interactions of OsGP1 were predicted using STRING (Fig. 3F), and the results showed that it was co-expressed with CESA1, TUBB8, and BC1L4 with scores of 0.646, 0.611, and 0.59, respectively. The analysis of the OsGP1 promoter using PlantCARE (Fig. 3G) revealed that in addition to the basic promoter elements TATA-box and CAAT-box, it contains the stress-responsive element STRE, the MYB and MYC elements involved in environmental adaptation, the ABRE element involved in abscisic acid (ABA) response, the light response element G-box, LTR involved in low-temperature response, TCA-element involved in salicylic acid (SA) response, HD-Zip 1 involved in differentiation of the palisade mesophyll cells, GCN4 motif involved in endosperm expression, and TGACG-motif involved in methyl jasmonate (MeJA) response. Among them, G-box at position −1953, TCA-element at position −556 and HD-Zip 1 at position −903 had the highest scores of 10, 9 and 8.5, respectively.

Figure 3 Bioinformatics analysis of OsGP1.

(A), Nucleotide sequence and deduced amino acid sequence of OsGP1 coding region. (B) Secondary structure prediction. Blue, α-helix (h); green, β-turn (t); red, extended strand (e); purple, random coil (c). (C) Tertiary structure prediction. The model was constructed with Q851X5_ORYSJ as the template. The Global Model Quality Estimate (GMQE) score is 0.62 and the colours represent different model confidence scores. (D) Prediction of conserved domains of OsGP1 protein. Red, signal peptide; blue, transmembrane region; purple, low complexity region. (E) Prediction of the signal peptide. The S-score is higher in the signal peptide region, the C-score is highest at the cleavage site, the Y-score is a parameter that comprehensively considers the S-score and C-score, and the Y-max score is the putative cleavage site. (F) Protein interaction relationship. The red node is OsGP1 (OsJ_13402), and the nodes in other colors represent the ten proteins predicted to interact with OsGP1. Different coloured lines represent different interaction types. Black, co-expression; purple, experimentally determined; yellow, textmining. (G) Analysis of OsGP1 promoter. All cis-acting elements selected for display have a PlantCARE matrix score ≥5.

A total of 43 protein sequences were aligned from Blastp (NCBI) to construct a phylogenetic tree (Fig. 4A). Four proteins in the same branch with OsGP1 were selected for multiple sequence alignment, including that the O. sativa Indica group protein OsI_14363 had 99.54% similarity with the OsGP1 from O. sativa japonica group studied here. The similarity of OsGP1 to mucin-1-like from Oryza brachyantha was 75.36%, while its similarity to GUJ93 from Zizania palustris was 69.61%. The results of Clustal Omega were imported into Jalview, and the analysis showed that most of the conserved regions of the protein were from the 171th amino acid site to the end of the protein (Fig. 4B).

Figure 4 Phylogenetic tree and multiple sequence alignment of OsGP1 protein.

(A) Phylogenetic tree. (B) Multiple sequence alignment of the sequences in the same branch with OsGP1. Sequences were aligned using Clustal Omega with default settings. The phylogenetic tree was constructed in MEGA6.0 by using the neighbour-joining (NJ) method (bootstrap value, 1,000).

OsGP1 localizes in the cell membrane and cell wall

The subcellular localization of OsGP1 was predicted using PSORT, and the results showed that the probability of OsGP1 protein localization in the cytoplasmic membrane was 46%, 28% in the endoplasmic reticulum membrane, and 10% in the endoplasmic reticulum (Table 2). To further confirm the localization of the OsGP1 protein in vivo, pGWB5-OsGP1 tagged with GFP was transiently transformed to the onion epidermal cells with a gene gun. Consistent with the predicted cytoplasmic membrane localization, the green fluorescence of the OsGP1-GFP fusion protein was found to be expressed in the cell membrane and cell wall (Fig. 5).

Table 2 Subcellular localization prediction of OsGP1.

Location	Probability	
Plasma membrane	0.460	
Endoplasmic reticulum membrane	0.280	
Endoplasmic reticulum	0.100	
Outside	0.100	
Note: The subcellular location of OsGP1 was predicted by PSORT (http://psort1.hgc.jp/form.html).

Figure 5 Subcellular localization of OsGP1-GFP fusion protein in onion epidermal cells.

GFP and OsGP1-GFP driven by 35S promoter under fluorescence, bright field, and merged views. Bar, 50 μm.

Identification of rice overexpressing OsGP1

The rice calli were infected and transformed with Agrobacterium tumefaciens EHA105 containing pGWB5-OsGP1 plasmid and differentiated on the selection medium containing hygromycin. The T0 generation plants were transplanted to pots and genotyped by PCR, and the results showed that OsGP1 was integrated into the genome of the T0 generation lines (Fig. 6A). Transgenic rice seeds were harvested and screened for two generations under hygromycin to obtain T3 generation seeds. Total RNA was extracted from T3 generation lines and reverse-transcribed into cDNA as a template, and expression of OsGP1 was detected in the transgenic rice seedlings by qRT-PCR (Fig. 6B). The results showed that the expression of #2 line was up to 11-fold higher than WT, and the overexpressing lines #4 and #5 expressed more than nine-fold of the WT. Therefore, the transgenic rice seedlings lines #2, #4, and #5 overexpressing OsGP1 were selected for subsequent resistance analysis.

Figure 6 Identification of OsGP1 transgenic rice lines.

(A) Identification of T0 generation transgenic rice. The integration of OsGP1 was detected by PCR using specific and vector primers. M, Marker, DL2000. WT, Wild-type line, Lj11; #1-#6, OsGP1 transgenic lines. (B) The expression level of OsGP1 in T3 generation transgenic rice were detected by qRT–PCR. The expression level of Lj11 was set to 1, and the Os18sRNA gene was used as an internal reference control. Data represent the mean ± SD of three replicates. Statistical analyses were performed using Student’s t test: **P ≤ 0.01.

Overexpression of OsGP1 enhances soda saline-alkali stress tolerance in rice

Three-leaf stage seedlings of overexpressing OsGP1 rice lines T3-#2, #4, and #5 were treated with different concentrations of SAE, and the growth phenotypes were observed after 7 days (Fig. 7A). With the increase in SAE content, injuries of rice seedlings gradually increased, the leaves change from green to yellow, and the full leaves shrivelled. The transgenic lines were found to have less damaged leaf than WT. In the treatment groups with H2O:SAE ratios of 3:1, the WT differed significantly from overexpression lines, with a higher number of dead plants. Fresh weight analysis showed that the overexpression lines have higher fresh weight than WT plants. Similarly, the overexpression lines were significantly taller than that of WT under stress (Figs. 7B, 7C). Furthermore, the MDA content increased in all lines with increase in SAE concentration, but the overexpression lines had significantly less MDA than WT (Fig. 7D). Also, antioxidant enzyme activity assay showed that overexpression lines had higher SOD and POD enzyme activities than WT. SOD activity was significantly different between WT and overexpression lines under 3:1 and 2:1 concentrations of H2O:SAE, while POD activity showed the greatest difference when the H2O:SAE ratio was 4:1 (Figs. 7E, 7F). These results suggested that the overexpression of OsGP1 has an effective protective function against plant injury under high pH and high concentrations of ions, and that the overexpression of OsGP1 improved saline-alkali tolerance in rice.

Figure 7 Tolerance analysis of rice overexpressing OsGP1 to soda saline-alkali stress.

The three-leaf stage overexpression lines T3-#2, #4, #5 and Lj11 seedlings were treated with different ratios of SAE for 7 days, and water as control. (A) Growth phenotypes. (B) Fresh weight of five seedlings. (C) Plant height. (D) MDA content. (E) SOD activity. (F) POD activity. Data show the mean ± SEM of three replicates. Statistical differences are labelled with different letters using Tukey test (P ≤ 0.05, one-way ANOVA).

Discussion

Rice is an important food crop in the world, and its yield is affected by soil salinity and alkalinity. Alkaline salt stress can inhibit the photosynthesis and growth of plants more than neutral salt stress. Under alkaline salt stress conditions, metal ions precipitates, thus decreasing the availability of nutrients in the soil (Guo et al., 2017). Under the interaction of high pH and salt ions, alkaline salts are more restrictive than neutral salts for the germination of seeds and the growth of seedlings (Wang et al., 2022c). When rice plants are subjected to saline-alkali stress, the plant first experiences several changes in the cell wall, including a reduction in cellulose content, disruption of pectin cross-linking, and accumulation of lignin, resulting in the inability of rice to grow normally (Liu et al., 2021). CWPs play an important role in plant defense against abiotic stresses. In the current study, OsGP1 was found to be a proline-rich protein containing a signal peptide and a transmembrane structure. The Locus ID of OsGP1 (Os03g0852400) in RAP-DB (https://rapdb.dna.affrc.go.jp/transcript/?name=Os03t0852400-02) labelled as OsAGP31, which was identified in a screen for AGPs in the rice genome. It belongs to the non-classical AGPs which contain an AGP-like region and other atypical regions, and was expressed in roots and panicles (Ma & Zhao, 2010). During the sexual reproduction of higher plants, haploid microspores divide through asymmetric mitosis producing a larger vegetative cell (VC) and a smaller generative cell (GC) (Lu et al., 2021). The GC divides further into the twin sperm cells (SCs) for double fertilization, whereas the VC exits the cell cycle and serves as a companion cell for the GC and its daughter cell SCs (Berger & Twell, 2011). The vegetative cell wall protein OsGP1 is expressed in panicles at the stages of megaspore and microspore development, and STRING interaction analysis showed that it is co-expressed with the tubulin protein TUBB8, indicating that it may play a role in rice reproductive development. The OsGP1 promoter sequence contains cis-acting elements involved in differentiation of the palisade mesophyll cells (HD-Zip 1), hormone regulation (ABA, SA, and MeJA) and stress response (MYB, MYC and LTR) (Wang, Chen & Li, 2019; Chen et al., 1995). HRGPs play an important role in plant biotic and abiotic stress responses, and generally have high expression in roots to enhance the mechanical strength of cell walls to withstand the mechanical force when roots penetrate the soil, high expression of the HRGPnt3 in tobacco during lateral root formation indicates the same function (Keller & Lamb, 1989; Velasquez et al., 2011). As a member of the HRGP family, the expression of OsGP1 in roots may be related to its function in resisting adversity. The interaction prediction results showed that OsGP1 has co-expression with several proteins including CESA1, TUBB8 and BC1L4, which are involved in cellulose synthesis, plant growth and development, hormone signalling regulation, and tubulin synthesis. Cellulose synthase gene 1 (CESA1) is required for the crystallization of β-1,4-glucan microfibrils, which is related to the main mechanism of cell wall formation (Burn et al., 2002). Mutants of CESA have lower cellulose content, and there is a negative correlation between cellulose content and plant growth under abiotic stress (Hori et al., 2020). OsCESA9/OsCESA9D387N heterozygous plants enhance plant resistance to salt stress by deregulating the toxicity of ROS, scavenging ROS, and indirectly affecting related genes such as OsCESA4 and OsCESA7 (Ye et al., 2021). TUBB8 has the same protein sequence as the β-tubulin protein OsTUB8 in japonica. OsTUB8 is mainly expressed in anthers and pollen and is an anther-specific microtubule protein that has a unique role in microtubule formation during anther, pollen development and pollen tube growth. Its expression is upregulated by gibberellin (GA3) and may be involved in GA-regulated anther and/or pollen development (Yoshikawa et al., 2003). Proteomic analysis of salt-adapted cells (A120) from Arabidopsis thaliana callus showed that compared with wild-type cells, the differentially expressed proteins in A120 cells were strongly associated with cell structure-associated clusters, including cytoskeleton and cell wall biogenesis. Genes such as TUB4, TUB7, and TUB9 were induced to be expressed in A120 cells, and the overexpression of Arabidopsis thaliana TUB9 gene in rice increased the tolerance to salt stress (Chun et al., 2021). OsBC1L4 is a plant-specific glycosylphosphatidylinositol (GPI)-anchored protein that is a key regulator of directional cell growth and cellulose crystallinity, and its mutation results in reduced secondary cell wall thickness and cellulose content. The promoter analysis, co-expression analysis and organ-specific expression of OsGP1 showed that it may respond to changes in the external environment and internal hormone levels in plants, and play a role in cell wall structure building to maintain cell morphology and protect plants from damage under stress. Subcellular localization showed that the OsGP1 protein was localized to the cell membrane and cell wall, reflecting the specific expression characteristics of cell wall proteins. Based on protein interaction and localization analysis, OsGP1 may be related to ion channels, cellulose synthesis, or the synthesis and secretion of fructose and other non-cellulosic polysaccharides. Virus infection triggers several inducible basal defense responses, and the HRGP family proteins EXTs are involved in plant cell wall reinforcement and defense. In potato virus Y (PVYNTN)-infected potatoes, the synthesis of EXTs is induced, whereas the synthesis of the catalytic subunit of cellulose synthase (CesA4) is reduced. The active trafficking of these proteins occurs as a step-in potato cell wall remodelling in response to PVYNTN infection (Otulak-Kozieł, Kozieł & Lockhart, 2018). Saline-alkali-tolerant rice exhibited higher germination rate, root length, shoot length, fresh weight, and dry weight than sensitive rice under saline-alkali stress. Transgenic rice plants overexpressing OsGP1 and Lj11 (WT) were treated with different ratios of SAE for saline-alkali stress, the results showed that compared with Lj11, the fresh weight and plant height of transgenic lines were higher, and the degree of leaf yellowing was lower, indicating that the overexpression of OsGP1 enhances plant resistance to saline-alkali stresses. The MDA content can reflect the degree of membrane lipid peroxidation, which is an important parameter to reflect the antioxidant capacity of plants (Gaweł et al., 2004). SOD and POD are key enzymes for ROS scavenging, and their high intracellular activity usually alleviates the damage of ROS and restores ROS homeostasis in plant cells. The MDA content of overexpressing OsGP1 lines is lower than that of Lj11, and the SOD and POD activities were significantly higher than those of Lj11 under stress. Thus, overexpressing OsGP1 rice cells scavenges the stress-induced ROS by increasing antioxidant enzyme activities to reduce cellular oxidative damage.

These results indicate that OsGP1 improves plant stress resistance by increasing the strength of plant cell wall to maintain the homeostasis of the intracellular environment. The current study provides experimental basis for analyzing the structure, function and stress resistance mechanism of rice vegetative cell wall proteins. The elucidation of the molecular function of OsGP1 in cell wall stress resistance and its possible role in plant reproduction and differentiation requires further research.

Conclusion

Functional characterization of the rice OsGP1 gene revealed that OsGP1 plays an essential role in soda salt-alkali stress. In overexpression lines, OsGP1 modulates ROS scavenging, cellular homeostasis and cell wall signaling and formation. Taken together, we conclude that OsGP1 is a unique gene with the potential to be used as a candidate gene in the molecular breeding of rice to achieve food security and rice tolerance to multiple abiotic stress.

Supplemental Information

Supplemental Information 1 Primers used in this study.

Supplemental Information 2 Recombination strategy for the Gateway method.

A, Gateway cloning technology used in this study. The construction strategy was created with Biorender (https://www.biorender.com/). B, Details of the linkage between OsGP1 and GFP proteins.

Supplemental Information 3 Raw Data.

We thank Dr. Qingyun Bu for providing Lj11 seeds.

Additional Information and Declarations

Competing Interests

Author Contributions

Data Availability

The authors declare that they have no competing interests.

Fengjin Zhu conceived and designed the experiments, authored or reviewed drafts of the article, and approved the final draft.

Huihui Cheng performed the experiments, authored or reviewed drafts of the article, and approved the final draft.

Jianan Guo analyzed the data, authored or reviewed drafts of the article, and approved the final draft.

Shuomeng Bai performed the experiments, prepared figures and/or tables, and approved the final draft.

Ziang Liu analyzed the data, prepared figures and/or tables, and approved the final draft.

Chunxi Huang analyzed the data, prepared figures and/or tables, and approved the final draft.

Jiayi Shen analyzed the data, prepared figures and/or tables, and approved the final draft.

Kai Wang performed the experiments, analyzed the data, authored or reviewed drafts of the article, and approved the final draft.

Chengjun Yang conceived and designed the experiments, authored or reviewed drafts of the article, and approved the final draft.

Qingjie Guan conceived and designed the experiments, authored or reviewed drafts of the article, and approved the final draft.

The following information was supplied regarding data availability:

The raw measurements are available in the Supplemental File.

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
