# Peer review of "Vegetative cell wall protein OsGP1 regulates cell wall mediated soda saline-alkali stress in rice"

_PeerJ, doi:10.7717/peerj.16790_

## Round 0.1 · original submission · Major Revisions

Two experts assessed your manuscript and found the content interesting and a contribution that fits well in this journal. However, there are some concerns that need to be addressed before acceptance for publication. There are two experimental issues raised by Reviewer 1 that I consider essential to be addressed, along with the form changes requested by Reviewer 2.

·

Basic reporting

Zhu and colleagues report the cloning, sequence, and analysis of OsGP1, a cell wall protein of unknown function. This work is interesting and relevant; it provides insight into the plant tolerance mechanism in alkaline and saline soil. The manuscript is written well.
In the following lines, I provide the observations and comments on this manuscript based on the manuscript line numbering on the review pdf document. All the comments done by this reviewer are in good faith and attempt to improve the manuscript's content.

Experimental design

After reading the manuscript, this reviewer thinks that as supplementary material, the details on the plasmid map and construction strategy should be provided (figure of the primers used and recombination strategy for the Gateway method, final plasmid map). Here, a particular aspect should be indicated, how the two proteins were fused? If no linker was incorporated between OsGP1 and the GFP protein, this could potentially create problems in protein location, especially on the subject this reviewer raised below since the authors present only one fluorescent cell.
Please add the map and construction strategy as a figure for plasmid pGWB5 (as supplementary material).
The authors should clearly state the statistics and programs used in the methods section. This reviewer thinks that the statistical analysis is correct, but a better interpretation could be achieved by using a Tukey-Kramer test; please consider this suggestion.

Validity of the findings

This reviewer thinks that the introduction lacks the information to fully grasp the rationale behind choosing OsGP1 for this study and why the strategy was chosen.
The section in lines 178-184 is confusing (for further detail, please see the Additional Comments section. Here, there is a discrepancy between the description in the introduction and these lines. Also, the gel provided is not useful without the proper labeling. The first gel should contain enough information regarding what it is. Also, in the gel, the lower band, I assume, is the primers or something else; I think authors should repeat this gel to provide a better image and label clearly what each band is. The same is true for the digested plasmid DNA; as stated in the Experimental Design section, a map is needed to assess the products that I, as a reader, should interpret and see.
The model presented in Figure 2 Panel C requires additional information regarding the quality of the model, and the template or templates used to model it, since the authors used Swiss model. Also, the bioinformatic analysis would be enriched if, in lines 195-197, the authors provided a functional role for the neighborhood of OsGP1 and the analysis scores since the authors are not providing a rationale for the use of STRING and how that information was further used. The promoter analysis also needs to be further supported by the score of each binding site for the transcription factors found in this promoter sequence.
Regarding the phylogenetic analysis and sequence comparison, this reviewer suggests that authors analyze the sequence of OsGP1 to assess putative glycosylation sites, reinforcing the homology to mucin-1 (doi.org/10.1093/glycob/cwj110). Also, authors are encouraged to perform a structural comparison of mucins with known structure; this reviewer finds that the presented structure here is not homologous to the known mucins in the PDB. Please revise this, and perhaps it should be included as a novel mucin fold.
Lines 211 to 214 need more context. To assess the location of OsGP1, a fusion with GFP was constructed… something like this. Also, the aim was to correlate in vivo the prediction of the subcellular localization of this protein. Additionally, the authors should provide a rationale for using transformed onion cells. This reviewer thinks that is evident for the thickness and transparency of the tissue, but this should be stated. Also, please state that this is a transient transfection.
The subcellular localization of the fusion protein seems weak; only one cell is shown. Authors should describe in the response letter indicating why this is the case. This reviewer thinks that more cells should be shown and provide strong evidence of the subcellular location of this protein.
As stated in Figure 1, the gel in Figure 5 panel B also requires a better presentation.
Regarding the stress results on transgenic plants, the results suggest that the plants are more resistant to stress. I recommend the authors provide a solid comparison of the results and show the fold difference with the wt plant to assess the results shown in Figure 6.
Finally, authors are encouraged to confirm the String neighborhood of OsGP1 and at least one or two examples of the stress response elements by measuring the expression rate in the Wt and transgenic plants obtained in this work. This is important for supporting the role of OsGP1 presented in this work.
Discussion and conclusions summarize the results; please revise and provide more insight on the actual mechanistic effect of OsGP1 than repeating the results.

Additional comments

Minor comments on the manuscript are listed here:
In line 44, I suggest using extreme alkaline conditions instead of high pH.
In line 55, I suggest biochemical and molecular mechanisms instead of biochemical reactions to cope…
Please revise the structure in lines 63-67, here the arrangement should be revised since the list of CWPs is, in the first instance, referred to Jamet et al., 2008 and then the authors provide the work of Calderan-Rodrigues et al., 2019 which provides a better assessment of this topic. Perhaps authors should consider citing this reference instead of Jamet’s reference or place as “previously, Jamet’s work considered CWP’s in this main functional role… and then, Calderan-Rodrigues et al. found that…” This suggestion is to clarify this paragraph.
In line 69, please clarify “pathogen infection”, I think the authors meant “pathogen entry” or “pathogen sensing and entry”; this is not clear to this reviewer.
Please italicize Arabidopsis in line 70.

·

Basic reporting

Some small writing errors and proper description of results should be taken care of. My suggestions in this regard are highlighted in the attached file.
The legends of the figures could be more descriptive, so that their interpretation would be easier.

Experimental design

Although adequately described and referred to in the methodology, the reading would be easier if in the description of the results of the Bioinformatic Analysis section reference was made to the databases consulted. For the rest, the experiments are adequately described and were carried out adequately.

Validity of the findings

The results are adequately supported. However, the conclusions would be more compelling if they were listed, rather than described in a continuous text.

Additional comments

It is a very common mistake that statistical probabilities are only referred to as "<" when the correct value is "≤" (equal to or less than).
The experimental strategy would be clearer if, as shown in Figure 5, a diagram of the cloning vector is included in Figure 1. The experimental strategy would be clearer if, as shown in Figure 5, a diagram of the cloning vector is included in Figure 1.

---

## Round 0.2 · Minor Revisions

Both Reviewers appreciated that the manuscript was improved but it still has room for improvement, as pointed out in their comments.

·

Basic reporting

I thank the authors for providing a revised version of the manuscript, I think that overall, the manuscript has improved greatly. In the following sections I provide minor issues that need to be addressed before the manuscript enters the next stage.

Experimental design

I have no further comments in this section.

Validity of the findings

One of the question I asked is regarding Figure 1 and S1, please add the residues linking the two proteins (OsGP1 and GFP), if no linker was added, indicate this in the methods section.
For the STRING Figure (Figure 3) in the legend, please indicate the meaning of the color of each connecting node. For non-expert readers this must be provided.
In the methods section, lines 185-187, please revise the writing, here is not mention that a Tukey Kramer test was done, but in Figure 7 is indicated. Please correct this.

Additional comments

I think authors should provide a deeper explanation on the disordered prediction of OsGP1 protein (Figure 3 and lines 198-204), I encourage authors to at least compare two modeling approaches besides SWISS-MODEL.

·

Basic reporting

The writing can/should be carefully reviewed, perhaps it would be advisable to use a professional writing service. In particular, the legends of the Figures must be rewritten to make them much more explicit.

Experimental design

The writing can/should be carefully reviewed, perhaps it would be advisable to use a professional writing service. In particular, the legends of the Figures must be rewritten to make them much more explicit.

Validity of the findings

The results are valid and experimentally supported, but unfortunately some are inconsistent with those included in the previous version, this should be carefully reviewed.

Additional comments

As can be seen in the attached tracked document, the manuscript was changed substantially. In fact, the writing has changed so much that it could well be considered a new submission. The title was changed, another author was included, and the writing was modified, adding, deleting, and editing texts from practically all sections. Evidently the changes were made with the intention of improving the report and in fact the discussion is better, more bibliographic sources (at least 9) were consulted (and included in the references section). However, I find some aspects of the results section very surprising and prevent me from accepting the manuscript for publication. In the current Figures 2 and 6 (1 and 5 in the previous version) the gels were changed, this is good because the current ones are of better quality, but it stands out that the sizes of the amplified fragments, particularly in Figure 6, are of different sizes. to those shown in the previous version. In this same sense, the interactomes shown in Figures 3 (current) and 2 (previous) are different, even though they were built with the same data and software.


I sincerely appreciate the efforts of the authors to meet the demands of the reviewers. Unfortunately, although the results and conclusions were essentially preserved, the new version has so many changes that it seems like a new submission that, like the previous version, must be improved before being accepted for publication.

---

## Round 0.3 · accepted · Accept

The authors addressed all the pending concerns.

From the Section Editor:

> I suggest carefully checking the writing. E.g. the final sentence of the abstract is somehow redundant "These results suggest that OsGP1 may be involved in stress response pathway, by enhancing cell wall mediated saline-alkali stress tolerance in rice." Also, in the Conclusions, I recommend reducing "filling phrases" such as "In the present study, we" and " The results presented here reveal .."